# 'From Dialogue to Action': The Promise and Challenge of a Multireligious Approach to Peacebuilding. A Myanmar Case Study

**Anna S. King * and Mark Owen**

Centre of Religion, Reconciliation and Peace, University of Winchester, Winchester SO22 4NR, UK;
mark.owen@winchester.ac.uk

* Correspondence: Anna.King@winchester.ac.uk

**Abstract:** This article explores whether a relational approach to peacebuilding, shared multireligious perspectives and widening networks can bring sources of strength which enable positive peacebuilding and create grassroots, cross-community peace. While religious peacebuilding organizations have become the object of a burgeoning literature, the role of multireligious organisations in peacebuilding has received far less attention. The purpose of this paper is to redress this lack. By examining the influence, challenges and benefits of multireligious approaches to transnational peacebuilding, we hope to develop a sharper and more critically nuanced understanding of the potential role of multireligious organisations in global peacebuilding, and consider what, if anything, distinguishes them from secular and other faith-based organisations. We do so by analysing the impact of a project carried out in Myanmar by Religions for Peace. The project provides three case studies which offer unique opportunities to consider the limits and potential of multireligious grassroots interventions in conflict contexts with very different histories and cultural configurations.

**Keywords:** religion; multireligious, peacebuilding; Religions for Peace; Myanmar; justpeace

## 1. Introduction

Peacebuilding scholars and practitioners have in recent decades advocated a more central role for religious peacebuilding, stressing religion's institutional capacity and spiritual resources for peace (Johnston and Sampson 1994; Sampson and Lederach 2000; Appleby 2000, 2015; Gopin 2000, 2002; Johnston 2003; Smock 2006). Today, there is far greater recognition of the potential contribution of faiths to peacebuilding, and of the ways in which they can be instrumentalised and mobilised. Increasingly, organizations and governments promote religion as a means to build global peace and security, while sectors that previously marginalised religion, regarding it primarily as a driver of conflict, now accept that it has an essential role to play in any integrated, multi-layered approach to peacebuilding.[1] Religious leaders are increasingly perceived as influential cultural brokers and interpretivists who can effectively promote humanitarian law and human rights (Cismas 2014).

There is a growing body of scholarly literature on religion and peacebuilding, but a huge diversity of ideas and approaches about what religious peacebuilding means. Religious non-profit organisations (NGOs) are a heterogeneous group of conservatives and progressives, international

---

[1]   E.g. Lederach 1997; Appleby 2000; Gopin 2000; Abu-Nimer 2003; Little 2007; Lederach and Appleby 2010. See also UNHCR 2014 Partnership note on faith-based organizations, local faith communities and faith leaders. Geneva, Switzerland.

and regional, small and big, old and young, and religious peacebuilding occurs in many forms, from shared humanitarian activities and peace and reconciliation efforts at the community level to long-term efforts to address structural injustice (Petersen 2010). This article focuses on transnational multireligious approaches, seeking to understand the ways, if any, in which their impact differs from those of monofaith or secular organisations which also work across cultural and religious boundaries and engage in interreligious dialogue and action as a mechanism for peacebuilding. Based on case studies from Myanmar, it analyses the implications and potential added value of multifaith-based peacebuilding for religious peacebuilding as a field of practice and contextualises research findings within current theoretical debates.

'Interdependence is built on relationships and relationships are the heart and bloodlines of peacebuilding' (Lederach n.d.). Relationship is at the heart of all peacebuilding and NGOs dedicated to peacebuilding, whether secular or faith-based, try to bring about social healing and interreligious or interethnic reconciliation. A few adopt a multireligious approach to peacebuilding, an approach explicitly founded on relational strategies, shared perspectives and widening networks. Such organisations have received little scholarly attention—due perhaps to the fact that they make up only a small percentage of all religious NGOs (Petersen 2010). We hypothesise that in some contexts, they offer unique advantages. They are intentionally and necessarily relational and pluralist, both institutionally and in their relations with the outside world, and may therefore avoid to some extent the dominant, Western Christian use of language about religion, and the 'significant ideological baggage, including Eurocentrism, technocentrism and historic connections with Empire' (Richmond and Mac Ginty 2015). By examining the influence, challenges and benefits of multireligious approaches, we hope to develop a sharper and more critically nuanced understanding of their potential role in global peacebuilding. We do so by analysing the impact of a project carried out in Myanmar by Religions for Peace International (R*f*P-I), in partnership with Religions for Peace Myanmar (R*f*P-M).

## 2. Religions for Peace: Different Religions, Shared Values

Religions for Peace describes itself as the world's largest and most representative multireligious coalition advancing common action among the world's religious communities for peace. The organisation comprises the World Council of senior religious leaders, six regional interreligious bodies, and more than eighty national ones, as well as the Global Women of Faith Network (WoFN) and Global Interfaith Youth Network (IFYN). Religions for Peace's approach is explicitly founded on relational strategies, shared perspectives and widening networks. Although R*f*P often uses the terms multireligious, interreligious and interfaith synonymously, its preference for the term 'multireligious' indicates that its central function is purposive collaborative humanitarian action. 'It works to transform violent conflict, advance human development, promote just and harmonious societies, and protect the earth.' [https://rfp.org/about/]. R*f*P's interpretation of 'religion' drives its goals, motivation and methods. A Religions for Peace Assembly Theme Paper (n.d.) states:

> Each religious tradition represented in RfP has its own positive vision of Peace, which includes its understanding of human dignity, individual and communal flourishing, the obligation to be in harmony with others and the natural world, and its notion of ultimate fulfilment. In RfP, each religion's positive vision of Peace is respected as being sincerely held by the believers of that religion. While great care is taken to avoid a "syncretistic" blending of the beliefs of diverse religions, RfP recognizes that diverse religious visions of Peace do provide the bases for carefully discerning elements of a positive, multi-religious vision of Peace. From its beginning, RfP has labored to discern and express elements of a shared positive vision of Peace. This is done by discerning and expressing consensus through shared values, rather than in terms of the differing doctrines that are unique to each religious tradition.

R*f*P's public statements conceptualise all major or world religions as bounded and discrete with distinctive histories, heritages and cultures, but as sharing common values and deserving equal

respect. Key assumptions are that all world religions share common values of peace, justice and compassion, that religions and religious leaders possess vast untapped resources to contribute to peaceful solutions to the world's crises, and that interreligious dialogue and multireligious action promote the reframing of attitudes towards 'the other.' Simultaneously, RfP perceives itself as developing a consciously chosen universal ethic with an expanding circle of moral concern. This combination of ethical universality and religious particularity is believed to enable affiliated religious leaders and communities to retain a sense of their own separate identities but to co-operate for the greater good. RfP claims that multireligious peacebuilding brings an added dimension to global peacebuilding and politics which emphasises relationship-building, concepts of justice, compassion and human flourishing. For the individual member, these values are affirmed through private prayer and the use of religious doctrine, storytelling and symbolism (Singh 2015). RfP tends to employ moments of silence in meetings, and relegate deep theological engagement, interritual participation, prayer, and symbolic content to the private devotional sphere (see Schwarz 2018, pp. 159–69 for an alternative view).

Religions for Peace International sponsors many peacebuilding initiatives which are then implemented by its national and regional affiliates. Religions for Peace Myanmar (*RfP-M*), one such affiliate, was founded in 2012, and claims to be the country's 'first representative and action-oriented interreligious body for reconciliation, peace and development.' It brought together members and organisations from Myanmar's historic 'four major religions,' Buddhism, Christianity, Islam and Hinduism, and included the Buddhist Sitagu Sayadaw community, the Ratana Metta Buddhist Organization, the Myanmar Council of Churches (MCC), the Catholic Church, the Hindu Community in Myanmar, and the Islamic Center of Myanmar. Dr. William F. Vendley, the then Secretary General, stated at its foundation that 'Communal harmony must be the bedrock of authentic development for Myanmar. The shared moral and spiritual values of Buddhism, Islam, Christianity and Hinduism advanced by *Religions for Peace* Myanmar will provide a basis for communal harmony, while its grassroots multi-religious projects will translate these values into needed action.'[2]

## 3. 'Multi-Religious Networks Promoting Religious Diversity and Tolerance'

### 3.1. Aims and Objectives

The RfP project, 'Multi-Religious Networks Promoting Religious Diversity and Tolerance,' is the main source of the data that inform this article. Its objectives were to build multireligious capacity to support peacebuilding by training religious leaders on conflict resolution and enhancing the capacity of local women of faith groups for conflict prevention and mediation; to promote social cohesion and reconciliation; to facilitate the conditions for the smooth return/resettlement of IDPs; and to strengthen RfP-M's national organisation and support the community of practice on multireligious peacebuilding.

The project proposal emphasised collective action and bridge-building. The councils (IC, WoFN and IYN) would be trained in contemporary peacebuilding and reconciliation skills by a process of cascading from international to national and then local level. Also central to the project was RfP's global campaign 'Welcoming the Other'[3] which promotes mutual respect, diversity, tolerance, understanding and acceptance. The local groups were to collaborate in civic activities which contributed to the common good as well as interfaith/interreligious dialogue. Different religious communities were twinned ('*Place of Worship Pairing*') to encourage religious and community leaders to build trusting and respectful interreligious relationships in preparation for 'welcoming the other.'

---

2  http://act.religionsforpeace.org/site/MessageViewer?dlv_id=8821&em_id=6301.0
3  The 9th World Assembly of Religions held in 2013 inaugurated this campaign, recommending ways in which 'Welcoming the Other,' would guide future actions. religionsforpeace.org/sites/default/files/publications/22 November Newsletter Day 3.pdf

'Congregations' would then be introduced through social interactions such as community dinners, exchange visits, open houses and sessions where each group 'shared' about their faith and customs. Religious leaders might be asked to give a spiritual talk on peace preceded by prayer as a way of familiarising guests with their religious practices or as a way of building bridges. At the national level, religious leaders were tasked with writing booklets setting out the peaceful teachings of each of Myanmar's major religions and engaging the media in alternative narratives challenging religious or nationalist hate speech. This proved critical because social media—and especially Facebook—played a key role in exacerbating fear and tensions in all three conflict areas (Kinseth 2018).

*3.2. The Evolving Data Collection and Evaluation Process*

Evaluation, predominantly qualitative, was incorporated into the project from the beginning and contributed to programme design and planning. R*f*P-M facilitated the authors' access to project staff and beneficiaries during the period 2015–2018. Their reports analysed the experiences of the multifaith peacebuilding groups and beneficiaries and explored the project's impact in motivating change. The 2015 initial conflict assessments seek to establish participants' ownership of the project and perceptions of the drivers of conflict. The mid-term evaluation (2017) focuses on the project's promotion of pluralism but also its limitations. The final report (2018) measures progress against the objectives and indicators defined in the funding proposal and analyses longitudinal levels of impact.

Data, both qualitative and quantitative, were collected through workshops, focus groups, interviews and participant observation. The authors interviewed staff, religious actors and project beneficiaries, and also held wider consultations with religious communities, administrative officials, aid workers, IDP camp organisers, military personnel, armed militias, ecological advocacy CSOs, etc. They also drew upon the reports compiled by staff as part of their own monitoring and evaluation schedule. These provided quantitative information about the project's reach, age, gender and religion of participants, the number of activities completed as well as qualitative observations and self-reports of attitudinal and behavioural change.

Peacebuilding is a complex, multifaceted process of change. On paper, a faith-based peacebuilding project can signify to its donors, beneficiaries, and stakeholders its aims and objectives, planned timelines, baseline studies, intended outcomes and outputs. The funding document imagined a linear process of reconciliation, requiring quantitative and qualitative measures of change. Participants and staff also imagined incremental progress towards social healing. However, as Lederach and Lederach point out (2010, p. 43), there are tensions between the experience-based perspective and the analytic metaphor structures found in the literature. In volatile situations where renewed cycles of violence and human rights violations are possible, reconciliation is always multifaceted and complex and depends on a mass of interacting and dynamic forces. It requires the building of relationships and trust which cannot always be measured in quantifiable ways.

The evolving and fluid nature of the peacebuilding dynamics was evident in the evaluation and data collection process which revealed the multitude of factors operating. During the early and mid-term evaluations staff and participants were focused on both the potential and limitations of the project design and implementation. However, by the time of the final evaluation, staff knew that future funding might well rest upon the outcome. Beneficiaries also wanted the project and funding to extend and were aware that the jobs of field monitors and Yangon staff were at risk. There was therefore a concerted effort to understand and reflect upon the project's positive achievements.

## 4. The Project Context

Myanmar was at the time of research (2015–2019) in the process of transitioning from decades of military rule and oppression, with the Aung San Suu Kyi-led National League for Democracy (NLD) achieving a landslide victory in the 2015 national elections. The NLD's victory was generally welcomed as heralding further progress towards democracy and economic development, based on a strategy of non-violence and national reconciliation. However, Ang San Suu Kyi's triumph was regarded with caution in several ethnic states because she had been silent about their rights, and

because, in the 2015 elections, the NLD did not put forward a single Muslim candidate. Renewed cycles of violence began to damage the new government's international reputation, threaten external investment and highlight continuing military control.[4] Since the constitution reserves 25 percent of all parliamentary seats and control of three important ministries—Defence, Border Affairs and Home Affairs—to military appointees, the country's military retained considerable political and economic power.[5] Equally significant factors in the conflicts were the failure of the NLD government to address the structural injustices in ethnic minority areas and the lack of political will to work with minority parties and organisations.

Myanmar is a religiously and ethnically diverse country, with close and complex relationships between religion and ethnicity. It has a Bamar/Burman and Buddhist majority, with a large Muslim minority and members of other religious traditions and ethnic groups (Walton and Hayward 2014). The challenge remains of fully reconciling Myanmar's diverse peoples and including them in one political system (Farelly 2014). Since independence, non-Bamar ethnic groups have demanded equal rights, equal opportunities and self-determination. Many resent the political and cultural dominance of the Bamar and have little confidence in the NLD or its national reconciliation process. Whilst global attention has been fixed on the Rohingya crisis, the decades-long offensives waged against non-Burman nationalities by the Burmese military continue to create major humanitarian crises. Under the NLD, two more ethnic armed groups have signed the Nationwide Ceasefire Agreement (NCA), which now has ten signatories.[6] However, the government has so far failed to bring peace, not only in western Rakhine but also in the northern Kachin and northeastern Shan states. Numerous other ethnic conflicts have fuelled claims of widespread and systematic human rights violations by the Burmese military and ethnic militias.[7]

The project was designed to respond to these political, religious and social tensions by nurturing grassroots peacebuilding, combining reliance on international and national leadership with an ethos focused on local empowerment. It emphasised the importance of multilevel processes and the value of identifying, drawing upon and building up existing networks and connections. It was directed by two senior members from the R*f*P international secretariat which oversees the organisation and works closely with various UN agencies and other religious and secular NGOs.[8] The project was based in Yangon, a cosmopolitan, multi-ethnic, multireligious city,[9] and the headquarters of R*f*P-M. It was steered by a Core Group of senior national religious leaders.[10] and implemented by a paid program manager, project co-ordinator and local field monitors. The project was intended to ameliorate ethnic

---

[4]　Report of the independent international fact-finding mission on Myanmar 12 September 2018. https://www.ohchr.org/Documents/HRBodies/HRCouncil/FFM-Myanmar/A_HRC_39_64.pdf.

[5]　This was documented by a UN Fact-Finding Mission on Myanmar which exposed military business ties, calls for targeted sanctions and arms embargoes.https://www.ohchr.org/EN/NewsEvents/Pages/DisplayNews.aspx?NewsID=24868&LangID=E.

[6]　See the reports of the 21st Century Panglong Peace Conference sessions. https://www.irrawaddy.com/news/burma/date-set-third-21st-panglong-conference.html.

[7]　On 11 December 2019, before the International Court of Justice, the Hague, Aung San Suu Kyi defended Myanmar against accusation of genocidal intent against the Rohingya and stated that there would be no tolerance of human rights violations in the Rakhine, or elsewhere in Myanmar. She argued that international law only gives international courts power to intervene when a country fails to prosecute crimes itself. https://www.statecounsellor.gov.mm/en/.

[8]　See Schwarz (2018, pp. 142–44) for details of how R*f*P draws on the language and agenda of the United Nations.

[9]　Even here religious tensions arose in the course of research, most notably perhaps in the murder of Ko Ni, a prominent Muslim lawyer in Yangon International Airport in 2017. In May 2019 a nationalist mob shut down Ramadan prayers. https://www.myanmar-now.org/en/news/nationalist-mob-shuts-down-ramadan-prayers-in-yangon.

[10]　They included Cardinal Charles Bo, Archbishop of Yangon; U Myint Swe, President, Ratana Metta Organization; Al Haj U Aye Lwin, Chief Convener, the Islamic Center of Myanmar; and Rev. Father Joseph Maung Win, Head of the Office of Yangon Archdiocesan Commission for Ecumenism and Interfaith.

and religious tensions in situations where identities, particularly minority identities, matter, and where people who do not share the majority identity or had outsider status could be physically displaced, expelled, 'resettled' or socially marginalised.

## 5. The Project Centres

It is beyond the remit of this paper to offer a full analysis of the conflict dynamics in each area, but extensive conflict analyses were conducted with partners in Myanmar to situate the research, map conflict issues, key stakeholders and peacebuilding resources. In all areas, respondents identified multiple competing forms of conflict, trauma and oppression. The Core Group were deeply mindful of security and the dangers to staff and participants. It was only after prolonged deliberation that Myitkyina, the capital city of Kachin State, Meiktila, a city in the central Mandalay Region, and Kyaukpyu, a major town in Rakhine State were selected as project bases. The original decision to set up a multireligious centre in Sittwe, the capital of Kachin State, was abandoned when orchestrated anti-Muslim violence made it too dangerous. In Meiktila and Myitkyina the interfaith councils were already established and many of the participating religious and civic leaders were known to each other and to the local field monitors who were solidly anchored in the local population with strong connections to established religious communities.

The conflict in Kachin has its roots in the unilateral abrogation of the Union of Burma constitution by the Ne Win regime in 1962 and is ostensibly an ethnic armed insurgency, with the Kachin Independence Army fighting the Bamar controlled military for the right to self-determination and independence. The Kachin, the dominant group, were evangelised in the early 20th century by American Baptist missionaries, and their modern identity is deeply interwoven with Christianity. Predominantly Baptist, they include a distinct Catholic minority.

In Meiktila religious tension has played a significant role in recent intercommunal violence and violations and is linked to the broader issue of Buddhist–Muslim relations in Myanmar. Meiktila is situated in the Mandalay/Sagaing region which has strong associations with Buddhist nationalism and the minority of extremist monks who have emerged as a political force becoming fierce defenders of Buddhist culture and way of life (Walton 2015, p. 17). In 2013, a dispute in a gold shop between the Muslim owner and two elderly Buddhist clients triggered a riot which spread to towns across Myanmar's central plains. Many people were killed, injured, or left homeless after widespread burning of properties.

Rakhine is one of the poorest of Myanmar states and many local Rakhine feel that they have an ongoing struggle to maintain Arakanese/Rakhine culture against both the Rohingya and the Central 'Bamar' Government. Unlike the Muslim Rohingya, the majority Buddhist Rakhine (or Arakanese) are officially recognized by the central government as an ethnic minority but feel marginalised in a country historically dominated by the Bamar. Rakhine ethnic minorities have fought for self-determination in Rakhine State since the early 1950s, and insurgent groups, such as the Arakan Army (AA), continue hostilities against the government. Indeed, many Rakhine believe that the Rohingya issue is instrumentalised by the national Government and army to control Rakhine, and anger is often directed as much against the Government and Tatmadaw as the Rohingya. While Rakhine Buddhists may be privileged in relation to non-Rakhine, they see themselves as equally subject to government repression, and, like other non-Bamar minorities, they demand autonomy under a federal set-up (Walton 2013). Having suffered decades of oppression and neglect, they are particularly receptive to political and religious propaganda, and tend to support the official line that the security forces are responding to a serious threat from Rohingya terrorists led by foreign Islamists. The complexity of the Rohingya crisis lies in the fact that Rohingya are not considered citizens of Myanmar, a fact that makes their case highly inflammatory (Kipgen 2013). In Kyaukpyu intercommunal violence erupted in June 2012; 147,000 people, including approximately 138,000 Rohingya, were internally displaced

and an entire Muslim neighbourhood burnt to the ground.[11]  During the entire duration of the project, the Kyaukpyu councils remained 'tri-faith' only (Buddhist, Christian and Hindu), partly because Kyaukpyu Muslims were interned in camps. This was a limitation with profound consequences.

**6. Principal Findings: 'From Negative to Positive Mind'**

The dynamics, complexities and interconnectedness of conflict requires an ongoing focus on building and strengthening the capacities of individuals, communities and organisations at grassroots level, and the opportunity to create cross-cutting linkages and relationships between practitioners throughout the world The findings showed that the Project's structural framework, religious and spiritual resources and local knowledge combined with a values-led, relational approach resulted in recognisable and sustained transformational change. In the following section the positive themes that emerged clearly and consistently from the mass of data collected are analysed.

*The project facilitated international, national and regional communication and support.* One of R*f*P's guiding principles is to link national, regional and international networks and structures. Communication and decision-making between and across the various levels was sometimes slow and difficult but enabled the International Project Directors and the Core Group to remain fully informed of what was going on at the local level. Simultaneously, international approaches to conflict transformation, peacebuilding and dialogue were disseminated down to the local groups, increasing the pool of trained peacebuilders able to conduct conflict prevention, mitigation and resolution activities within their own communities.

*The project enabled new relationships and friendships across religious and ethnic divides* which modelled positive interreligious relationships for the wider society. Responses in questionnaires and interviews indicate that the activities and trainings helped to transform attitudes from 'negative' to 'positive' mind, a phrase used by R*f*P members to describe on the one hand expanding empathy and communication across cultural, ethnic, religious and social divides and, on the other hand, the transformation of problematic senses of community identity, based on fear, exclusion, superiority over and demonization of the other. Typical remarks are: 'R*f*P is a model for Myanmar.' 'It brings people with different perspectives together.'[12]  The IC and WoFN members report feeling more able to challenge Facebook rumours and hate speech.[13]  Interview and questionnaire findings indicate that participants felt less threatened, fearful and suspicious, and that meeting people of different religions had made them feel more informed and more open-minded. 'We have become accustomed to visiting different religious centres and eating and drinking with people we previously passed by in the bazaars.'[14]

*The project enhanced the capacity building of the Women of Faith Network.* This was possibly its most significant outcome. R*f*P meetings provided a unique opportunity for women from different religious, ethnic, economic and educational backgrounds to find a new collective power. The religious dimension gave women a legitimacy that other peacebuilding initiatives could not. While the importance of the role of women in peacebuilding is increasingly recognised (Hayward 2015; Hayward and Marshall 2015), R*f*P from its foundation has always sought to empower women through the women's interfaith networks and its training programmes and support. Women involved in peacebuilding tend to gravitate to efforts that entail sustained interfaith and intrafaith

---

[11]　The Myanmar military response to the ARSA attacks of 2017 led to one of the worst humanitarian crises of the 21st century. An estimated 25,000 Rohingya Muslims have been killed and over 700,000 have fled to neighbouring Bangladesh. The UN report of August 2018 accused Myanmar's military of genocide. https://www.ohchr.org/EN/HRBodies/HRC/MyanmarFFM/Pages/Index.aspx.

[12]　Notes of meeting of Myitkyina IC and WoF, April 2017.

[13]　Since the outbreak of the Rohingya refugee crisis in August 2017, U.N. investigators and human rights groups have criticized Facebook harshly over alleged lack of action against accounts that used its platform to encourage violence against the disfranchised Rohingya Muslims.

[14]　Notes from interviews with Kyaukpyu councils in January, February and March 2018.

relationship building, particularly in traditionally patriarchal and deeply religious societies like Myanmar where religious leaders are still venerated and influential as a source of ethical guidance. In this project women consistently report feeling empowered by engaging in social action and the training for leadership. They comment that they drew inspiration and support from religious sources, teachings and interfaith dialogues, but also from the RfP national (female) mentors. The women's groups, more than the men's, were vibrant, enthusiastic, and keen to develop their respective networks, knowledge and skills. IC (male) leaders were often heads of local religious communities (monastics, priests, imams) or professionals (doctors, lawyers, teachers, etc.), who led busy lives outside the family. Women seized the opportunity to go beyond the confines of family and community, giving many illustrations of ways in which they had benefitted from the trainings on conflict resolution, dialogue facilitation and 'Welcoming the Other.' They reported that they had gained greater respect within their own communities, and that some had developed strong leadership roles within the wider society. A response by a Kyaukpyu Christian young woman is typical, 'I enjoyed pairing with Buddhists. I made new relationships and friends. I can dare talk in society and am more confident.'[15]

*The project helped participants ignore power imbalances and elevated the status of minority religious groups.* In Kyaukpyu, for example, Hindu respondents reported that while previously they were made to feel inferior, they were now given greater respect. The Hindu priest remarked happily that now members from other faiths gave him their phone numbers and visited the temple. 'They became very close. Their mindset changed.'[16] Individual Rakhine Buddhists also admitted to feeling more positive about the Hindu community. One young man stated that whereas before he had no desire to mix with 'Kalar' [an insulting term], he now had Hindu friends. The Rohingya continued to be spoken of as illegal migrants, landgrabbers and potential terrorists,[17] but the seeds of reconciliation were sown.[18]

*The project created awareness of shared values and religious difference.* Participation in dialogue sessions and interreligious celebrations and ceremonies was regarded as a positive development leading to explicit recognition of an essential shared humanity and greater understanding of ethnic and religious differences. The President of the All Burma Sikh Religious Council, a Myitkyina member of IC, commented that the project had promoted greater intercommunity cooperation and awareness of Sikh identity and teachings, while the dynamic president of the Gorka-Hindu Hindu Women's Association (a member of WoFN) became accepted by all religious communities as a respected mediator and facilitator. Members of all faiths attended the Interfaith Community Prayer Service in Myitkyina for assassinated NLD Legal Adviser Lawyer U Ko Ni and taxi driver U Nay Win. Non-Muslims reported that they would never have dreamt of going but for the attitudinal changes wrought by RfP. In Meiktila, there were small but telling acts of kindness. A Muslim woman who had stopped selling her halal chickens in the market after the riots restarted her business with the support of RfP Buddhist leaders. In Kyaukpyu, young people explained that they had not studied religion at school and that RfP activities had brought them

---

[15]  Interview with the Christian community, April 2017.

[16]  Interview with the pujari and members of the Hindu community, May 2017.

[17]  Fear often drives hostility. Remarks by a Rakhine Buddhist IDP woman were typical. 'I am frightened of Muslims. Bangladesh is a small country. It can't support its people. Muslims come over the border. Three people go on a fishing trip and six return. They [Muslims] have villages on the border. Rakhine people are frightened to live there. They [Muslims] demand land from the Government. The NGOs and INGOs are sympathetic only to Rohingya.' Buddhist IDP Camp, May 2017.

[18]  Some made the journey from opponent to advocate. Ma Soe Aung was among those Buddhists who lost everything in the Kyaukpyu riots and forced to live in an IDP Camp for almost five years. When IC members came to the camp to identify concerns and discuss plans for resettlement, she listened to the interfaith discussions and eventually joined WoFN, becoming a strong advocate for the civic rights and benefits of all people, including the Rohingya. Her name has been changed for security reasons.

understanding of different religious traditions, more friends and greater confidence in exploring religious themes.

*The project challenged stereotypes and power structures.* Feedback analysis showed that participants believed that increased knowledge and dialogue were effective in deconstructing stereotypes, and that their new understandings were transferred to families and communities. Among Kyaukpyu members there was increasing recognition that some of the concerns about the Rohingya were distorted and that 'Bengalis' had come to be defined by their 'otherness.' Interfaith discussions and settings reminded people of the time before the riots when they lived in peace with their neighbours, 'before Rohingya began to make demands for citizenship.' Discussions became notably more searching and less emotional and incendiary. Young people challenged hostile, demeaning and dehumanising stereotypes, remembering past friendships with Rohingya schoolfellows and neighbours. Local people began to greet Rohingya IDPs when they saw them shopping in the market and some volunteered to attend dialogues in the 'Muslim' camp. In the final dialogue sessions in 2018, IC and WoFN members spoke movingly and prayerfully of their desire for peace and the possibility of future coexistence.

*The project trained local people from all religious/ethnic backgrounds in contemporary peacebuilding skills.* Members were sensitised to the complexity of religious identities, providing them with basic skills enabling them to promote conflict transformation and peacebuilding within their own communities and in multireligious settings. They were also trained to assess rumours and deconstruct the fake news and propaganda which dehumanises neighbours as enemies.[19] This had observable results. When Meiktila communities felt the effects of the worsening situation in Rakhine and fears of religious violence spread, local R*f*P councils began to share accurate updates with each other and with their religious institutions and networks, while the IYN started a social media campaign to combat sensational rumours and promote positive messages.

*The project participants contributed to the wellbeing of IDPs.* Decisions over the return/resettlement of IDPs/refugees were made by central or state governments. However, R*f*P councils liaised with local government officials and initiated dialogues with IDPs in order to facilitate their smooth return. Myitkyina is ringed with IDP camps—with shelter ranging from permanent woven bamboo huts to makeshift tarpaulin tents. Here, R*f*P councils helped to ensure the efficient running of the camps, mediated between the camps and host communities and supported female heads of families, widows, children and the elderly. In Meiktila, they gave substantive practical help to resettled Muslim and Buddhist families, while in Kyaukpyu, staff encouraged support and donations to IDPs in both the 'Buddhist' and 'Muslim' camps.

## 7. The Challenges and Limitations of Multireligious Peacebuilding

While these positive themes are clearly shown in the data, there were simultaneous and circular themes that did not fit a strictly linear metaphor of progression and transformation. In each locality there were stories of displacement and experience of violence. Lived experience for many was often of structural violence or decades of armed conflict. In Kachin, many communities had family members who had been killed, drug addiction was rampant and unemployment high. Fighting was never far away. In Meiktila, individual Muslim R*f*P members confided privately that they felt like second-class citizens and feared further attacks. In Kyaukpyu, both the Rohingya and Rakhine faced uncertainty and the possibility of renewed violence.

*International universalism versus local experience.* R*f*P's global message that all religions are essentially religions of peace, assumes a culture of peace, dialogue and forgiveness. Whilst this can inspire hope, it can also lead to the avoidance of complex, difficult topics. For example, R*f*P international and national support for the UN's investigation and report into atrocities in Rakhine, contrasted with its angry rejection at local level. It is particularly significant then that one of R*f*P-M's founder members, Al Haj U Aye Lwin, became a member of the UN's Advisory Commission on

---

19      Owen and King 2018.

Rakhine State chaired by Kofi Annan. Rakhine Buddhists, on the other hand, felt misunderstood, ignored and demonised by the scale of international support for the victimised Rohingya, and angry about illegal migration, encroachment on Rakhine land, the high birth rate among Muslim families and the threat to Rakhine and Buddhist culture. They were also afraid of revenge attacks should Rohingya refugees return. There is therefore among many Rakhine deep-seated frustration, anxiety and sense of injustice (Leider 2017). Meanwhile, the Rohingya IDPs in the Kyaukpyu camp, concerned about their children's future, repeatedly spoke of their desire to go home. The crisis encapsulated a complex set of humanitarian issues—notably questions of internal displacement and resettlement, the contested status of citizenship of a large part of the Muslim population, deep political mistrust that divides the Buddhist and Muslim communities, and ongoing communal tensions that threaten peacebuilding

   *'Othering' the out-group: societal, religious and 'nationalist' pressures to conform.* Conflict often heightens conformity to group identity, and it requires great personal courage to defy social pressures and norms. Individuals and societies form their identities through complex and enduring processes of differentiating ('othering') and integration (Tajfel and Turner 1986). In Kyaukpyu *RfP* meetings, there was a danger that R*f*P meetings could bring unity by creating a common enemy, and that the minority Christian and Hindu communities would make common cause with Rakhine Buddhists by 'othering' the Rohingya. Christians felt that open support or sympathy for Rohingya would make them a target not only of their own more hard-line community, but of Buddhist extremists. The Hindu community spoke of double jeopardy.[20] They believed that ARSA terrorists had already killed 300 Hindus because they identified them as Buddhist sympathisers,[21] but they also felt at risk from widely shared anti-Muslim, anti-'Bengali' sentiments. Rakhine Buddhists feared the reactions of the wider Bamar community and Arakan nationalists if they appeared sympathetic to the Rohingya.[22] A Buddhist teenager remarked, 'If I speak out in favour of the Rohingya I will be mocked as a 'jihadi bride.'[23]

   *Multireligious peacebuilding and development*. R*f*P staff with professional experience of development projects reported that religious peacebuilding was far more demanding in that it could not appeal directly to the communities' primary interests which were economic.[24] They pointed out that poverty, poor education, unemployment, and the culture of drugs all fuel religious violence and ethnic tensions, and that the project, albeit unintentionally, often excluded the poorest and those at most risk from violence. R*f*P councils tried to remedy this by urging that the project budget should include costs for travel to and from meetings and suggesting that multireligious vocational courses or co-operative business ventures would attract a wider range of participants. Such proposals fell outside the funding remit. The young people interviewed also found many of the activities irrelevant to their future lives and prospects. R*f*P-M later responded to these challenges in policy statements, stating that the provision of interreligious vocational training or mentoring, together with facilities for study and IT, would be a more strategic and effective way to bring religious communities together.[25]

---

[20]  The Hindu community was said to number 250 people and to have good relations with the Rakhine Buddhists. Interviewees said that Indians came in the time of the British—some were soldiers, many were from Bengal. Some have citizenship papers, others do not. Most leave school at the age of 14 or so and take up work as carpenters, tailors and motorbike taxi drivers (interviews in Kyaukpyu, November 2015).

[21]  Amnesty International reports that, 'A Rohingya armed group brandishing guns and swords killed around 100 Hindu people in Myanmar's Rakhine State, ahead of the violent ethnic cleansing carried out by Myanmar's security forces last year.' https://www.amnesty.org.uk/myanmar-crisis-rohingya-armed-groups-massacred-hindus

[22]  For this reason, R*f*P staff decided in 2019 to take local Rakhine leaders to Yangon, where they were able to talk more freely (conversation with Patrick Aung, the Program Manager, 22 June 2019, Singapore).

[23]  Interview with IYN, Kyaukpyu, February 2018.

[24]  Interview with project co-ordinator, Yangon, April 2018.

[25]  Reports of the R*f*P Advisory Forum on National Reconciliation and Peace in Myanmar for November 2018 and May 2019.

*Religious leaders and influencers.* Staff often found the recruitment and retention of local religious or civic leaders a major challenge. Core members who are often Yangon-based have little time for local involvement, while at grassroots level, some religious/civic leaders were deterred from engaging practically and ideologically with a multireligious project. Much of this complexity is because religion is not only about faith, values and spirituality, but a marker of social, ethnic and national identity, shaped by shared history, mythology, culture, and sense of destiny. As a marker of individual and cultural identity, religion in Myanmar can be as much about power and privilege as doctrine or sacred text.[26] In all three conflict areas, there were assertive influencers who claimed to protect their religious heritage, and whose aggressive use of religious symbols in public crystallised a kind of xenophobia. We interviewed Baptist KIO leaders in Kachin who were convinced that Jesus was on their side fighting for justice, and Buddhist political activists who argued that the Tatmadaw's campaign against the Rohingya protected Myanmar against regional Islamicisation and the kind of Islamist terrorism experienced internationally. And while multireligious organisations attract 'liberal' leaders sympathetic to multifaith peacebuilding and dialogue, in times of intercommunal tension feelings of fear and insecurity operate to silence many.

*The politics of identity.* Religious, ethnic and political identities are often blurred, and whilst religious leaders should by their nature 'lead,' their role is also to represent their community (Sampson 1997). In Myanmar, anxiety about the fragility of Buddhist culture is intensified by globalised imaginaries of endangered identities (Gravers 2015). R*f*P leaders often wrestle with conflicting and ambivalent desires to serve and protect their own community while engaging in interfaith mutuality and relationship building. Even unintentionally religious values and language may be used by the same leaders both to justify violence and exclusion, and to offer an ethical and spiritual critique of violence and the depersonalisation of the other. The relation between the state and the Buddhist sangha is confused and controversial. Some local monks interviewed lamented the decline of the moral authority of the Sangha in peacebuilding. Others believed that monastics are principally charged with guarding their own tradition and its privileges, or that monks should not be involved in political, religious or social activism. Buddhist religious leaders are often reluctant to criticise militant monks like Ashin Wirathu,[27] and may perceive multireligious co-operation and advocacy as diluting and relativizing the national cultural heritage. Even non-partisan engagement can be misunderstood. In Meiktila, Buddhist monastics felt constrained to avoid interfaith events after a local newspaper published a picture showing them sitting as equals among leaders from other religious communities. This created a wave of popular protest, particularly on social media—from then on, Buddhist monks either sat apart or did not attend.

*International projects versus local contexts.* R*f*P is action driven, and the project-based approach to complex change, the urgency of the timescale and short-term funding meant that sometimes the instrumental need to satisfy donors took precedence over the goal of producing self-sustaining communities of religious activists and the construction of a free civil society. This became particularly apparent when key staff resigned and there was an interim before new (excellent) staff were recruited. There was then great pressure to complete the backlog of required activities before the project's closing deadline. Activities were often generic or shaped by R*f*P's international policies. For example, tree planting, a popular activity, fitted in with R*f*P-I's global advocacy to reduce climate change. R*f*P councils' suggestions for activities of fundamental concern to local communities frequently fell outside the project's funding criteria; they included interfaith peace education in schools and colleges, multireligious co-operatives, support for orphanages, HIV/AIDS help centres, out of school children, campaigns against women and drug trafficking and support for the elderly.

---

[26]    Grant (2004, p. 272) argues similarly that in the Irish context, these sets of meaning are often blurred, confused or even at odds, and the slippage between them gives rise to a kaleidoscopic variety of nuanced meanings.

[27]    The leader of the anti-Muslim movement in Myanmar who is accused of inciting riots and of using racism and rumours to spread hatred.

## 8. National Impact

One of the main goals of the project was to strengthen the national faith-based organisation (FBO) and to enhance its national profile. There was strong media coverage of the concluding conference in Yangon in 2018 which shared lessons learnt and good practice. It not only attracted R*f*P's international, national and regional religious/civic leaders and stakeholders but government officials, intergovernmental bodies, diplomats, civil society organizations, human rights activists and religious communities. And while the authors presented a careful overview of the main research findings, the project's success was also celebrated through personal narratives, music, poetry and the sharing of food.[28]

Religions for Peace-M began to engage politically at the national level, informed by its regional initiatives and supported by its international connections. *A Letter to the Peoples of Myanmar: Multireligious Vision of Peace and Development* was drawn up by an R*f*P-M delegation convened in Yangon and Nay Pyi Taw between 22 and 25 May 2018. This open letter presents a vision of peace based upon Myanmar's great religious traditions and a vision of development built upon the notion of human dignity, human rights and shared well-being. It calls for an international conference with the participation of concerned States, United Nations, The Association of Southeast Asian Nations (ASEAN), The International Committee of the Red Cross (ICRC) and other relevant actors to address the critical humanitarian issues facing Myanmar. On 25 May 2018, the *Letter* was delivered to Ang San Suu Kyi. She welcomed the delegation and emphasised the critical role of religious leaders in reminding their faithful of peace and loving kindness and leading them to action and working together. She also praised the future steps planned for continuing dialogue and multireligious cooperation.[29] The R*f*P delegation then travelled to Sittwe and Maungdaw in the north of Rakhine State.[30] And while the NLD government has continued to deny access to Rakhine to international investigators, and Aung San Suu Kyi has sought to contextualise the violence in Rakhine state as a civil war between the Burmese military and armed militia groups such as the Arakan Rohingya Salvation Army, the delegates were able to see from the air hundreds of burned and destroyed Muslim villages and to listen to first-hand accounts of mass murder and rape perpetrated by Myanmar's military. They urged the Union Government to take full responsibilities for a thorough and transparent investigation into multiple crimes perpetrated in Rakhine State, and encouraged a process to establish facts, restore rights of victims, and support social and political change.[31]

The inaugural R*f*P Advisory Forum on National Reconciliation and Peace in Myanmar took place in Nay Pyi Taw, on 21–22 November 2018, bringing together representatives of the Myanmar government, the military, parliamentarians from ruling and opposition parties, UN agencies, ASEAN, ICRC, international NGOs, national NGOs, religious leaders, and experts. It was intended to create "open space" for all sectors in Myanmar— 'to earnestly seek together a common path for peace.' In the opening ceremony, Aung San Suu Kyi praised the role of Myanmar's religious communities in convening the Forum, and said that 'placing emphasis on interfaith dialogue as a path to peace underscores the vital and indispensable role that the religious leaders play in shaping

---

[28]  The popular Burmese reggae singer Saw Pho Khwar sang peace songs, while moving stories of brave activists were told to loud applause. Stories like that of Ma Hto Hto Mar who fled her home during the violence in Meiktila in 2013 and lived in an IDP camp for two years. On leaving the camp, she joined R*f*P-M's Meiktila WoFN and became involved in intercommunal initiatives to rebuild relationships between Muslims and Buddhists. She was the first Buddhist woman to fight against Ma Ba Ta on behalf of a Muslim family.

[29]  https://rfp.org/religions-for-peace-multi-religious-delegation-meets-with-daw-aung-san-suu-kyi-to-deliver-the-letter-to-the-peoples-of-myanmar/).

[30]  Areas where ARSA attacks in August 2017 had triggered massive operations by security personnel on Muslim communities.

[31]  https://rfp.org/press-release-religions-for-peace-%E2%80%8B-multi-religious-delegations-visit-to-rakhine-state/.

a peaceful world.'[32]  The Forum called for an immediate cessation of hostilities nationwide, urged the government to seek accountability by carrying out an independent and impartial investigation of atrocities and human rights violations and supported a planned voluntary and safe return of verified refugees from Bangladesh with proper protection and accompaniment by international agencies such as UNHCR, the UN Refugee Agency. *RfP*-M also strengthened its collaborative efforts not only with the international body but other *RfP* affiliates. In particular, it liaised with *RfP* Bangladesh to provide humanitarian assistance to Rohingya Muslim and Hindu refugees.

## 9. Cultural Imperialism, Multiple Frameworks and the World Religions Paradigm

Questions of cultural privilege and political power arise whenever faith-based organisations work transculturally. In the introduction, we hypothesised that in some contexts multireligious peacebuilding is less at risk than monofaith organisations of proselytism, Eurocentrism, technocentrism and historical connections with colonialism and imperialism. However, questions remain about how far multireligious theories and practices undermine, resist, or supplement the hegemonic domination of certain forms of thought, understanding, and practice. *RfP* adopts a world religions paradigm in which the integrity of each religion is taken for granted. This paradigm, originally designed to emancipate the study of religion from its Christian confines, has been enthusiastically adopted in public discourse globally. However, its critics argue that such a perspective is an expression of modern Western hegemony which imposes a particular understanding of religion upon very diverse traditions (Gregg and Scholefield 2015; Owen 2011; Masuzawa 2005); Orsi 2002, 1997.[33]  Others maintain that such a perspective does conceptual violence to the nature of cultural worlds where a creedal or institutional affiliation to only one tradition is not the norm. The concept of 'religion' is then shaped by Christian expectations regarding religion, namely, creedal affirmation, sacred-secular contrast, and exclusive membership (Sharma 2011). This homogenises and essentialises religions and neglects the uncomfortable facts that most religious traditions have complex and contradictory histories and have at times been associated with institutions of oppression rather than peaceful co-existence, with their leaders provoking as well as resolving conflicts.

Despite such critiques, the world religions model remains an established way for many faith-based peacebuilders in increasingly plural and globalised societies to communicate, and while *RfP* public statements may essentialise religion away from its lived reality, they also function to inspire and encourage pluralist and humane visions of society and to create a new generation of reconciling leaders from diverse and dynamic backgrounds. During the project, we observed that, at all levels, religious leaders were routinely negotiating their way in the context of the embodied, relational, everyday lives of actors and communities, mindful that their main task was to keep alive and functioning the networks of relationships which create the goodwill, consent and voluntary participation of religious communities. At the local level, staff adopted coping strategies of non-partisan religious engagement, emphasising the project's humanitarian and civic benefits. Whilst from a sociological or anthropological perspective, 'religions' are far more fluid, internally contested and mutually shaping than *RfP*'s global statements suggest, the dynamic tensions between *RfP*'s global understandings and value commitments and its focus on local autonomy and empowerment prepare people to be resilient in the face of change. They can open up a range of opportunities for creativity and freedom that enable *RfP* to work towards greater effectiveness as a peacebuilding organisation (Schwarz 2018, p. 169).

---

[32]  https://rfp.org/press-release-religions-for-peace-advisory-forum-provides-open-space.

[33]  Cotter and Robertson (2016); McGuire (2008); Ammerman (2007); Orsi (2003, 1997); Hall (1997) question top–down authority and prioritise the everyday, relational, embodied and living nature of religion as a category and of the identity construction of religious actors and religious communities. Masuzawa (2005) argues that European Universalism has been preserved in the language of pluralism. Orsi (2003, p. 172) sums up much contemporary thinking, 'Religion is always religion-in-action, religion in-relationships between people, between the ways the world is, and the way people imagine or want it to be.'

## 10. Multireligious Peacebuilding, Justice and Human Rights

Any multireligious liberal approach focused on interreligious action and dialogue alone becomes complicit with power if it ignores extreme forms of violence, exclusion, and suffering, or fails to get to grips with the underlying problems of the broader conflicts—conflicts between state and central government, and between and within, ethnic 'nationalities' and religious groups (see also Omer 2015, Frewer 2017). Many international reports have highlighted the gross human rights violations in Myanmar and the need to strengthen accountability mechanisms, particularly as regards the military, and to address racial and ethnic discrimination.[34] R*f*P-M is well placed to combat intolerance and discrimination based on religion and ethnicity and to explore how religion relates to political, cultural and structural forms of violence. However, local issues of injustice, violation of human rights and long-term violence were confronted by R*f*P national leaders very cautiously given the sensitivities and dangers involved. Field monitors had neither the training nor the influence to address or challenge longstanding injustice. Yet the case studies' findings were encouraging. They suggest that it is only by attending carefully to the interpretations of all stakeholders that peacebuilding can begin to address the fundamental structural and political issues that give rise to conflict, and which may necessitate a systemic transformation of relationships in the region's political, economic and social policies and ethos. Therefore, if sustainable peace and coexistence in Myanmar is to be advanced, arguments for religious pluralism must be complemented by a series of political, economic, and legal reforms to address underlying insecurities and long-standing inequalities between communities.

It is tempting to argue that conflicts in Myanmar cannot be transformed without solving the deep-rooted structural factors that drive violence such as poverty, discriminatory governance, lack of education and employment. Yet Taylor (2015) remarks, 'What Myanmar needs is less ethnicized politics and more bottom-up integrative approaches towards the multiethnic complexity of the country.' By working in local contexts and providing opportunities and incentives for civic peacebuilding, the project encouraged communities concerned above all with their own economic, social, educational, and political issues to engage together in active citizenship for the common good. Interrelational, interpersonal multireligious projects commit participants to seeing that 'the other' has moral standing, and that 'the other' is in fact one of 'us.' They can provide safe, welcoming, ungendered spaces in which experiences of power and exclusion can gradually be healed and support collaborative notions of citizenship that transcend differences of nationality, gender, ethnicity and religion.

## 11. A Multireligious, 'Secular' and Values-led Model of Transcultural Peacebuilding

The project provided interesting material for one of the central questions in religious peacebuilding today—the extent to which faith-based initiatives conform with, contradict and extend conventional 'liberal peace' wisdom. Many scholars and practitioners dealing with peacebuilding, development and humanitarian aid speak of a 'turn' to the 'local', the religious and the elicitive, a move which makes the liberal approach appear short term, top down and insensitive to local interpretations and knowledge.[35] Within the religious peacebuilding discipline, this has often translated into a radical polarization of the secular liberal peace and religious justpeace. Our research in Myanmar and elsewhere has led us to question this religious–secular dichotomy and analytical framework (see also Gopin 2015; Little 2015; Chandler 2010; Clarke et al. 2008). In this project, R*f*P

---

34 See https://www.icj.org/myanmar-reverse-laws-and-practices-that-perpetuate-military-impunity-new-icj-report/; https://www.icj.org/law-and-policy-reform-necessary-to-combat-intolerance-and-discrimination-based-on-religion-or-belief-recommends-new-briefing-paper-on-myanmar/; https://www.icj.org/four-immediate-reforms-to-strengthen-the-myanmar-national-human-rights-commission/; https://www.icj.org/myanmar-reverse-laws-and-practices-that-perpetuate-military-impunity-new-icj-report/

35 See Mac Ginty and Richmond 2013' Mac Ginty 2018.

members worked to empower local ownership and participation but were also able to escalate problems to the national and even global stage, utilising three forms of capital: local capital, religious capital (institutional resources), and spiritual capital (ethical, theological, scriptural and spiritual values). However, R*f*P values are also aligned with so-called enlightenment or liberal principles based on an international order of rational, democratic, secular, free market and human rights abiding states. R*f*P normative statements on religion, its links with the UN and employment of professional staff, training, guides and methodologies compares well to the professionalized work of secular CSOs.[36] At the international and national levels, the organisation cooperates with the UN, and governments and partners in civil society on global challenges such as war and peace, sustainable development, climate change, extreme poverty and interfaith dialogue. Critics might regard this multireligious approach as simply an extension of the liberal peace. We found that, in practice, it seems to foster civic consciousness by integrating the values, expertise and global reach of religious traditions with the secular human rights agenda which guarantees equal treatment under the law, and supports democracy, justice, universal education, environmentalism and gender equality.

## 12. Reflections

While Religions for Peace's model of peacebuilding may be hybridic, its core membership is composed of religious activists from diverse religious and spiritual traditions who come together to seek ways in which harmony and peace can be established, and suffering and loss healed. Although this article is limited to a restricted examination of one organisation, our tentative and provisional conclusion is that multireligious peacebuilding has distinctive characteristics which in certain contexts support negotiation, reconciliation and resilience, and that facilitative transnational approaches to peacebuilding and development can be energising and productive compared to top–down advisory approaches or projects which take violent extremism as a central framework (Abu-Nimer 2003). However, the challenges are immense. Multireligious peacebuilding in armed conflict involves an attempt to build inclusive, resilient relationships at the local level, model non-violent debate on sensitive ethnic and religious issues and forms of collaborative humanitarian values which can be firmly embedded within local culture. Multireligious organisations must manage religious diversity, particularly in countries where religion or ethnicity are among the drivers of conflict. They must consider questions of appropriate forms of religious representation, and how far the spiritual resources of each religion should be used in the process of peacebuilding. In civil wars or military controlled states, its ethics often come into stark contrast with the pragmatic realities on the ground.

Nevertheless, the multireligious, international and multilevel nature of the project examined here contributed to its success in inspiring confidence in people still subject to stringent military and government control. It not only facilitated close engagement with key religious actors, but enabled funding, support and resources to be transferred across cultural and national boundaries. Such collaboration opens many doors. Peacebuilding, which includes all sides of the conflict, supports the research requirement not only to understand the uniqueness and multidimensionality of each conflict, but also the wider webs of meaning in which it is enmeshed and the complexity of the cultural, religious, ethnic, economic and historical layers of significance surrounding each. At times of heightened tension interreligious peacebuilding may strengthen democratic culture by cultivating a dialogue of reasoned ethical challenge.

Multifaith peacebuilding as a structured, accountable attempt to connect and sustain groups of people in creative, empowering and inspiring ways, has the potential to cut through and challenge communal complexities and destructive interrelationships. A broader variety of social interactions is more likely to create civic awareness and friendship than theological dialogue alone. The physical practice of shared action, whether humanitarian in order to benefit the community, or intercultural

---

[36]　The General Secretary was at this time an advisor to several governments on matters related to religion and peace. (https://rfp.org/connect/international-secretariat/; https://rfp.org/about/leadership/world-council/dr-william-f-vendley/)

in order to better understand the other, increases resilience and is productive of attitudinal change over time. Multireligious organisations are less likely to be accused of evangelism, and enable communities often defined by religious and ethnic identities to come to together in solidarity on a common platform of civic wellbeing and peacebuilding. The fact that local people then plan and manage public humanitarian and religious activities may engage the suspicions of local Government, police and army officials but is far less threatening than direct political action. Interfaith meetings also increase religious literacy, introduce shared terms and concepts, and present an opportunity for positive religious values like compassion, non-violence, forgiveness and hospitality to drive action.

The project was relatively small scale, yet it fulfilled its aim of strengthening peacebuilding capacity at the local level and contributing to R*f*P-M's political impact in national reconciliation and peacebuilding initiatives. By building networks between the national and regional centres, the flow of information, training and communication acted to bring the influence of international and national leaders to the regions and vice versa. These networks allowed participants to realise that they have an active and reflexive role in shaping, negotiating and changing their own beliefs and practices. It is partly the carefully gathered information on regional peacebuilding and conflict that allows Religions for Peace Myanmar to engage as a critical partner in the national peace process, to collaborate with political leaders and to present a liberal and progressive agenda based on preventing cycles of conflict, ensuring the human rights of ethnic and religious minorities, supporting women's leadership in promoting peace, and empowering the younger generations.[37] The sustainability of this model of peacebuilding is shown by the continuing presence of the regional councils and the national organisation's success in gaining funding to extend the project elsewhere.[38] In the national context, in Myanmar, R*f*P-M enables discussion of how religions are utilised for ideological and political ends and provides a platform for progressive leaders to support each other in confronting and challenging injustice and oppression. Multireligious peacebuilding is therefore an approach which can offer people a vision of a future beyond the violent conflict in which they are caught up. Participants in the project expressed this as a new way of seeing things, a change of heart and a renewal of hope.

**Author Contributions:** Both Professor Anna King and Dr Mark Owen contributed to the writing of this article. The first draft was written by Anna King and received critical comment by Mark Owen. Conceptualization, Anna King; methodology, Anna King and Mark Owen; software, Anna King and Mark Owen; validation, Anna King and Mark Owen; formal analysis, Anna King; investigation, Anna King and Mark Owen; resources, Anna King and Mark Owen; data curation, Anna King and Mark Owen; writing—original draft preparation, Anna King; writing—review and editing, Anna King and Mark Owen; visualization, Anna King; supervision, Anna King and Mark Owen; project administration, Anna King and Mark Owen; funding acquisition, Anna King and Mark Owen. Both authors have read and agreed to the published version of the manuscript.

**Funding:** Some data were collected as part of the evaluation of the Religions for Peace Project, 'Multi-Religious Networks Promoting Religious Diversity and Tolerance,' a peacebuilding initiative funded by the US Government Bureau of Democracy, Human Rights, and Labor. The authors received payment for the assessments carried out for Religions for Peace International, but not for their own research.

**Conflicts of Interest**: Professor Anna King and Dr Mark Owen declare a possible conflict of interests in that they supported Religions for Peace's peacebuilding project in Myanmar and acted as external evaluators. However, the funders had no role in the design of the study; in the collection, analyses, or interpretation of data; in the writing of the manuscript, or in the decision to publish the results. Dr Owen is the Secretary General of the European Council of Religious Leaders, a Religions for Peace affiliate in Europe. The work contributing to this article was undertaken in his professional capacity as Director of the Centre of Religion, Reconciliation and

---

[37]  https://rfp.org/press-release-religions-for-peace-advisory-forum-provides-open-space-

[38]  The Project's regional councils became self-sustaining while RfP-M gained further funding from the Norwegian and German Governments to continue the project elsewhere, including in the key towns of Sittwe, Mrauk U and Pyay.

Peace, University of Winchester, and was as far as possible an objective and fair assessment of the project's achievements measured against pre-determined indicators and outcomes.

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
