# Peer review of "The Promise and Challenge of Multireligious Peacebuilding in the 21st Century: A Myanmar Case Study"

_religions, doi:10.3390/rel11030121_

Round 1

Reviewer 1 Report

I really appreciated reading a paper with this topic, yet found it at many points a bit puzzling. That might be due to its style which seems to resemble a report more than an academic article in a few chapters. Furthermore, the text is a bit sloppy sometimes. That concerns citation of websites and the logical connection between some paragraphs and sentences.

But my biggest concern is due to its resemblence to a report. In order to get rid of that and thereby improve the scientific soundness, every chapter has to be embedded within an overall structure of argumentation which is grounded on a theoretic foundation. At the same time, the - indeed very intersting - findings of the project must be related to the theoretic foundation of the paper which is the multireligious approach to peacebuilding. I'm aware that the key points are already in the text, yet are not really obvious due to the great amount of mentioned empirical findings.

Reviewer 2 Report

The paper  “An Exploration of the Potentialities and Challenges of Multireligious Approaches to Peacebuilding: A Myanmar Case Study” aims to develop a sharper and more critically nuanced understanding of the potential role of multireligious organizations in global peacebuilding and to identify what distinguishes these organizations from secular or other  faith-based organizations.  The study is based on a three-level evaluation of the programs of Religions for Peace in Myanmar (RfP-M) in the period 2015-2018. While we have seen a keen interest in the role of religion in peacebuilding of the past few years, the authors argues that the peacebuilding of multireligious organizations have been under researched. The paper is well written and presents a solid analysis and interesting and important findings, there are however a few shortcomings that should be addressed before the paper is ready to be published in Religions.

The research is based on a careful evaluation of the programs that RfP-M. However, the presentation of the aims of the program and the data collection and research methods are presented quite late in the paper, and the reader is kept waiting for too long to understand the interesting material that the paper is built on. I suggest that the three sections under the headline “Multi-religious Networks Promoting Religious Diversity and tolerance” (from 2006 and onwards) are moved up, earlier in the paper, before the presentation of the project. Alternatively there should be a few sentences early on about the data collection and research methods.

The many conflicts in Myanmar is only superficially referred to on page 5, it is important for the sake of the paper’s argument that it is clear that the many conflicts are politically rooted. It is the structural injustices between the Bamar ethnic majority and the many ethnic minorities, dating back to the struggle for independence, that are still fueling the conflicts. Even the Rohingya issue is fundamentally political, although religion has become a factor that is used and mobilized on from all sides. This is a highly important point to present clearly in the paper, not least since it exemplifies the critique that the authors presents towards the end. When the authors state that “local issues of injustice, violation of human rights and long-term violence were confronted by RfP national leaders very cautiously” this should be a central point. There is much important healing and peacebuilding in these multireligious programs, as the authors exemplifies very nicely. However, the programs have their limits and will not help solve the conflicts, precicely because of the political and structural injestices that fuel the conflict. This is a very important point that needsto be more clearly argued for bot in the beginning and in the reflections towards the end.

I find that there is a general problem with many of the references in the paper. I expect that the footnotes should give a reference to support the claims in the text. However, the foot notes often don’t provide these references, but instead merely presents an extra point or piece of information. This is the case in for example footnote 27, but it happens quite frequently elsewhere as well and is a bit puzzling to the reader. I suggest that the authors refer to their rich data material more diligently throughout.

Lastly, a few minor points:

Title. Would it change the meaning if you switched “potentialities” with “potential”? it seems unnecessarily complicated as it stands now

129-131: While the NLD certainly is powerless in many respects due to the continued prerogatives of the military, it is too simplistic to claim that this is the reasons for renewed cycles of violence and the damaged reputation of the government. It is important to also state the failures of the NLD government to address the structural injustices in ethnic minority areas, and the lack of political will from NLD to work with minority parties and organizations.

Footnote 8. ALA is not actively fighting in Rakhine today. The Rakhine insurgency is now taken over by AA.

180 There’s also a considerable Catholic population among the Kachin

190 “to maintain Arakanese/Rakhine culture against both WHAT THEY CLAIM ARE ‘Bengali‘ Muslim migrants, and …” It should be clear that the term “Bengali Muslim migrant” is something that is claimed by the Rakhine

191-192. I don’t understand what the first part of this sentence.

195-197. This claim needs a reference. This is very much debated. Many Rakhine feel that the army used the threat from Rohingya militants to militarize the region to control Rakhine. The hatred against the Bamar military is in many instances more fierce than the hatred towards the Rohingya. Not least in the last two-three years.  

Round 2

Reviewer 1 Report

The paper's methodology and theoretic background are represented much clearer now. I have no further objections or comments.